# Cardiomyocyte-specific disruption of the circadian BMAL1–REV-ERBα/β regulatory network impacts distinct miRNA species in the murine heart

Mary N. Latimer [1,6], Lamario J. Williams[1,6], Gobinath Shanmugan[1], Bryce J. Carpenter [2], Mitchell A. Lazar [3,4], Pieterjan Dierickx [2,5] & Martin E. Young [1✉]

Circadian disruption increases cardiovascular disease (CVD) risk, through poorly understood mechanisms. Given that small RNA species are critical modulators of cardiac physiology/pathology, we sought to determine the extent to which cardiomyocyte circadian clock (CCC) disruption impacts cardiac small RNA species. Accordingly, we collected hearts from cardiomyocyte-specific Bmal1 knockout (CBK; a model of CCC disruption) and littermate control (CON) mice at multiple times of the day, followed by small RNA-seq. The data reveal 47 differentially expressed miRNAs species in CBK hearts. Subsequent bioinformatic analyses predict that differentially expressed miRNA species in CBK hearts influence processes such as circadian rhythmicity, cellular signaling, and metabolism. Of the induced miRNAs in CBK hearts, 7 are predicted to be targeted by the transcriptional repressors REV-ERBα/β (integral circadian clock components that are directly regulated by BMAL1). Similar to CBK hearts, cardiomyocyte-specific Rev-erbα/β double knockout (CM-RevDKO) mouse hearts exhibit increased let-7c-1-3p, miR-23b-5p, miR-139-3p, miR-5123, and miR-7068-3p levels. Importantly, 19 putative targets of these 5 miRNAs are commonly repressed in CBK and CM-RevDKO heart (of which 16 are targeted by let-7c-1-3p). These observations suggest that disruption of the circadian BMAL1–REV-ERBα/β regulatory network in the heart induces distinct miRNAs, whose mRNA targets impact critical cellular functions.

[1] Division of Cardiovascular Disease, Department of Medicine, University of Alabama at Birmingham, Birmingham, AL, USA. [2] Max Planck Institute for Heart and Lung Research, Bad Nauheim, Germany. [3] Institute for Diabetes, Obesity, and Metabolism, Perelman School of Medicine at the University of Pennsylvania, Philadelphia, PA, USA. [4] Division of Endocrinology, Diabetes, and Metabolism, Department of Medicine, Perelman School of Medicine at the University of Pennsylvania, Philadelphia, PA, USA. [5] German Centre for Cardiovascular Research (DZHK), Partner Site Rhine-Main, Bad Nauheim, Germany. [6] These authors contributed equally: Mary N. Latimer, Lamario J. Williams. ✉email: meyoung@uab.edu

Life on Earth has evolved to temporally compartmentalize biological processes at numerous levels of resolution. For the cardiovascular system, temporal governance spans from milliseconds (e.g., electrophysiologic parameters) to months/years (e.g., development)[1]. Regarding time-of-day, diurnal oscillations in heart rate, blood pressure, and cardiac contractility are well-established, driven in part by fluctuations in neurohumoral stimuli associated with daily behavior changes, such as sleeping/wakefulness and fasting/feeding[2–8]. At a biological level, daily fluctuations in signaling (e.g., phosphorylation status of signal transduction cascade components), metabolism (e.g., catabolic/anabolic pathways), and electrophysiology (e.g., excitation-contraction coupling) illustrate the dynamic and adaptable nature of the cardiovascular system[9] Adverse cardiovascular events (e.g., arrhythmias, sudden cardiac death, myocardial infarction, stroke) similarly exhibit daily fluctuations (with increased incidence in the early morning hours)[2,10]. Moreover, alterations in normal biological rhythms, secondary to genetics (e.g., single nucleotide polymorphisms in circadian-related genes), behaviors (e.g., shift work), and/or clinical states (e.g., sleep apnea) are invariably associated with increased cardiovascular disease (CVD) risk and mortality[11–14]. Despite this wealth of knowledge, the mechanisms leading to increased CVD risk follow circadian disruption remain poorly understood.

A cell-autonomous timekeeping mechanism, known as the circadian clock, has emerged as an intrinsic temporal orchestrator of cellular processes, within the 24 h timescale[15]. It has been estimated that this transcriptionally-based mechanism regulates between 3–16% of an organ's transcriptome[16]. For example, the cardiomyocyte circadian clock controls 6–10% of the cardiac transcriptome; many of these clock-controlled transcripts encode for proteins with known functions involved in signaling, metabolism, and electrophysiology[17,18]. Interestingly, comparison of hepatic transcriptomic and proteomic data revealed that approximately half of hepatic proteins exhibiting 24 h oscillations are not secondary to fluctuations in the corresponding mRNA, suggesting involvement of posttranscriptional mechanisms[19]. This may include small non-coding RNA species, which are established epigenetic regulators of biological processes through modulation of translation (as well as mRNA stability)[20]. An important class of small RNA species are microRNA (miRNAs) molecules; oligonucleotides of ~22 bases in length that form complexes with target mRNA species, attenuating mRNA stability and/or translatability[20]. A growing body of evidence indicates existence of an inter-dependent relationship between miRNAs and circadian clocks, whereby circadian clocks modulate miRNA levels over a 24 h period and miRNAs form feedback loops targeting specific circadian clock components[21,22]. In addition, miRNAs may serve as a functional clock output, potentially forming mechanistic links between circadian clocks and specific cellular processes[23,24]. However, despite increased appreciation that miRNAs are potent modulators of numerous cardiac processes[25], prior studies have not investigated circadian governance of miRNAs in the heart.

Multiple animal models of shift work recapitulate increased cardiovascular disease susceptibility[9]. For example, repetitive reversal of the light/dark cycle (twice weekly) results in adverse cardiac remodeling (exemplified by increased fibrosis, hypertrophy, and molecular markers of dysfunction)[26]. Similarly, murine models of cardiomyocyte circadian clock ablation develop age-onset adverse cardiac remodeling and cardiomyopathy[17,18,27]. Two such models include cardiomyocyte-specific Bmal1 knockout (CBK) and cardiomyocyte-specific Rev-erbα/β double knockout (CM-RevDKO) mice[18,27]. The precise mechanisms leading to cardiac pathology in these models are currently unknown. The purpose of this study was to determine

the extent to which cardiomyocyte circadian clock disruption impacts miRNAs in the murine heart. Initially, hearts were collected at distinct times of the day from 12wk old control and CBK mice (an age at which pathology has not yet developed), followed by unbiased assessment of miRNA species through RNA-seq. A 2-way ANOVA revealed 47 differentially expressed miRNA species in CBK hearts (in the absence of significant time-of-day-dependent effects). Following stringent cut-offs, bioinformatic analyses predicted that differentially expressed miRNA species in CBK hearts potentially influence processes such as circadian rhythmicity, cellular signaling, and metabolism. Of the miRNAs that were induced in CBK hearts, 7 were predicted to be targeted by the transcriptional repressors REV-ERBα/β (integral circadian clock components that are directly regulated by BMAL1, and exhibit low expression levels in CBK hearts). Similar to CBK hearts, CM-RevDKO hearts exhibited increased levels in let-7c-1-3p, miR-23b-5p, miR-139-3p, miR-5123, and miR-7068-3p. These observations suggest that disruption of the circadian BMAL1–REV-ERBα/β regulatory network in the heart leads to induction of a subset of miRNAs, whose predicted mRNA targets have established roles in processes that are critical for maintenance of cardiac function.

## Results

**Impact of cardiomyocyte-specific BMAL1 knockout on cardiac circadian clock gene expression**. To investigate whether genetic disruption of the cardiomyocyte circadian clock impacts levels of small RNA species in the heart, cardiomyocyte-specific BMAL1 knockout (i.e., CBK) mice were initially utilized. Previously published studies have confirmed disruption of the circadian clock only in the heart of CBK mice[18,28]. Accordingly, CBK and littermate control (CON) hearts were isolated at 4 h intervals over the 24 h day. Consistent with previously published studies[18], BMAL1 protein levels are decreased by 52.9% at ZT0 in CBK hearts (relative to CON hearts; Fig. 1a). We have previously reported attenuated/abolished 24 h rhythms in *bmal1*, *rev-erbα*, and *dbp* mRNA levels in these samples[29]; to characterize further the extent of clock disruption in these CBK hearts, additional core clock components (*cry1*, *cry2*, *per1*, *per3*) and clock-controlled genes (*hlf*, *tef*) were investigated at the mRNA level. A 2-way ANOVA revealed significant genotype main effects for all mRNA species investigated; *cry1* and *cry2* mRNA were significantly increased, while *per1*, *per3*, *hlf*, and *tef* mRNA were significantly decreased, in CBK hearts (independent of the time-of-day; Fig. 1b, c). The 2-way ANOVA also revealed significant time main effects for all mRNA species (Fig. 1b, c). Subsequent cosinor analysis indicated significant 24 h rhythms for all mRNA species in CON hearts (Supplemental Table 1). In CBK hearts, 24 h rhythms were significant only for *cry1*, *cry2*, *per3*, and *hlf* mRNA; of these gene, the amplitudes of *cry2*, *per3*, and *hlf* rhythms were significantly decreased in CBK hearts (relative to CON hearts; Supplemental Table 1). Collectively, these data confirm impaired circadian clock function in CBK hearts.

**Differential expression of small RNA species in CBK hearts**. We have previously reported the impact of cardiomyocyte BMAL1 deletion on the cardiac transcriptome (i.e., mRNA species)[18,28,30]. However, this analysis did not include small RNA species. To correct this knowledge gap, 'small RNA-seq' was performed on hearts collected from CBK and littermate CON mice at 4 h intervals over a 24 h period. A total of 4141 small RNA species were detected through this analysis, of which 1929 had an average expression of greater than 10 counts; the latter subset of small RNA species were utilized for subsequent data analyses. Somewhat surprisingly, the 2-way ANOVA revealed no

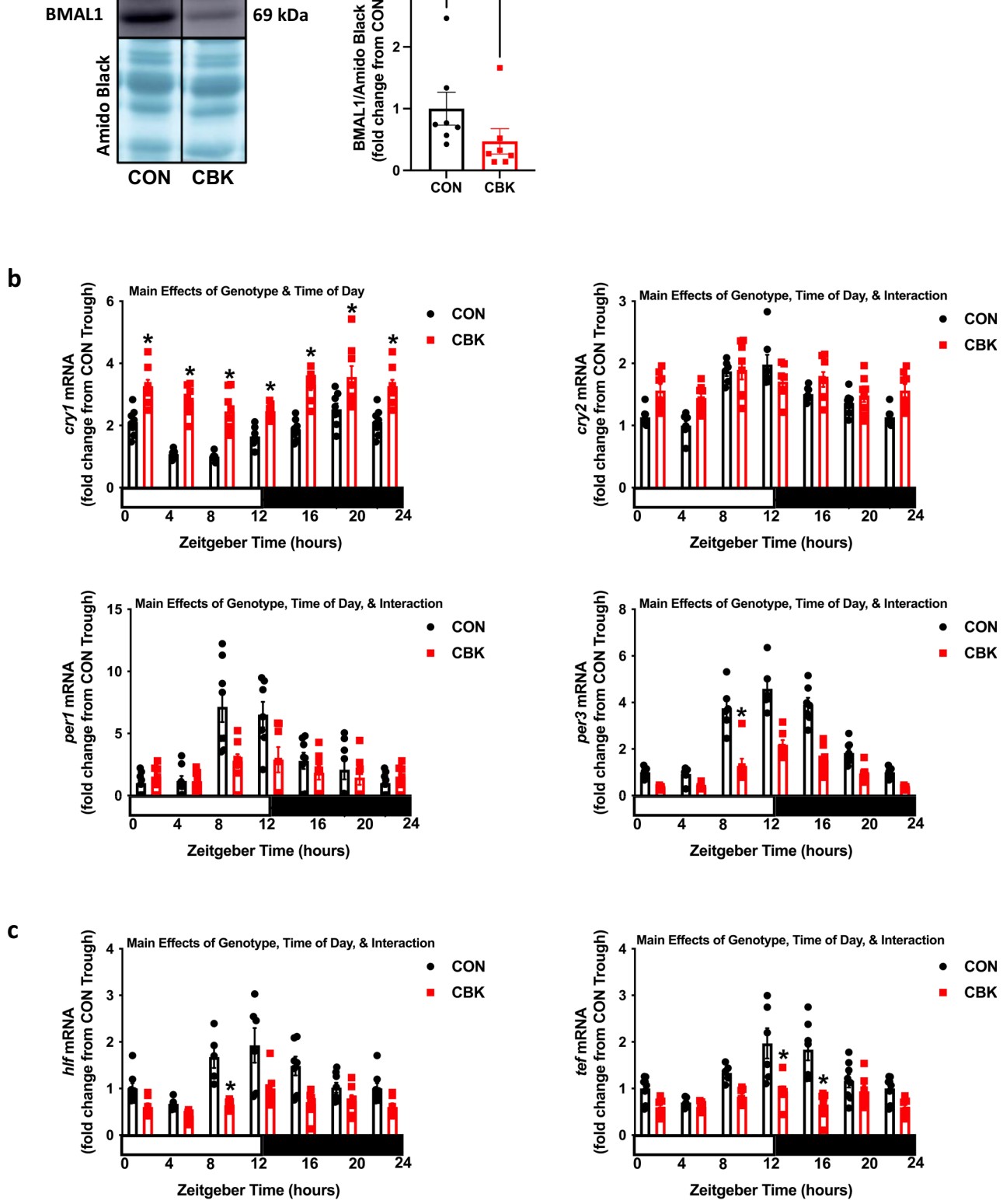

**Fig. 1 Cardiomyocyte-specific BMAL1 knockout (CBK) disrupts the circadian clock in the heart. a** Western blot of BMAL1 protein levels in CBK and littermate control (CON) hearts collected from 12wk old mice at ZT0 (*n* = 7). **b** Diurnal variations in *cry1*, *cry2*, *per1*, and *per3* mRNA levels for CBK and CON hearts collected at six distinct times of the day (*n* = 6–8). **c** Diurnal variations in *hlf* and *tef* mRNA levels for CBK and CON hearts collected at six distinct times of the day (*n* = 5–8). Data are presented as mean ± SEM. Main effects of genotype, time-of-day, and/or interaction are reported at the top of the figure panels. p-values are either reported as numbers, or as symbols: *$p < 0.05$ for CBK versus CON hearts.

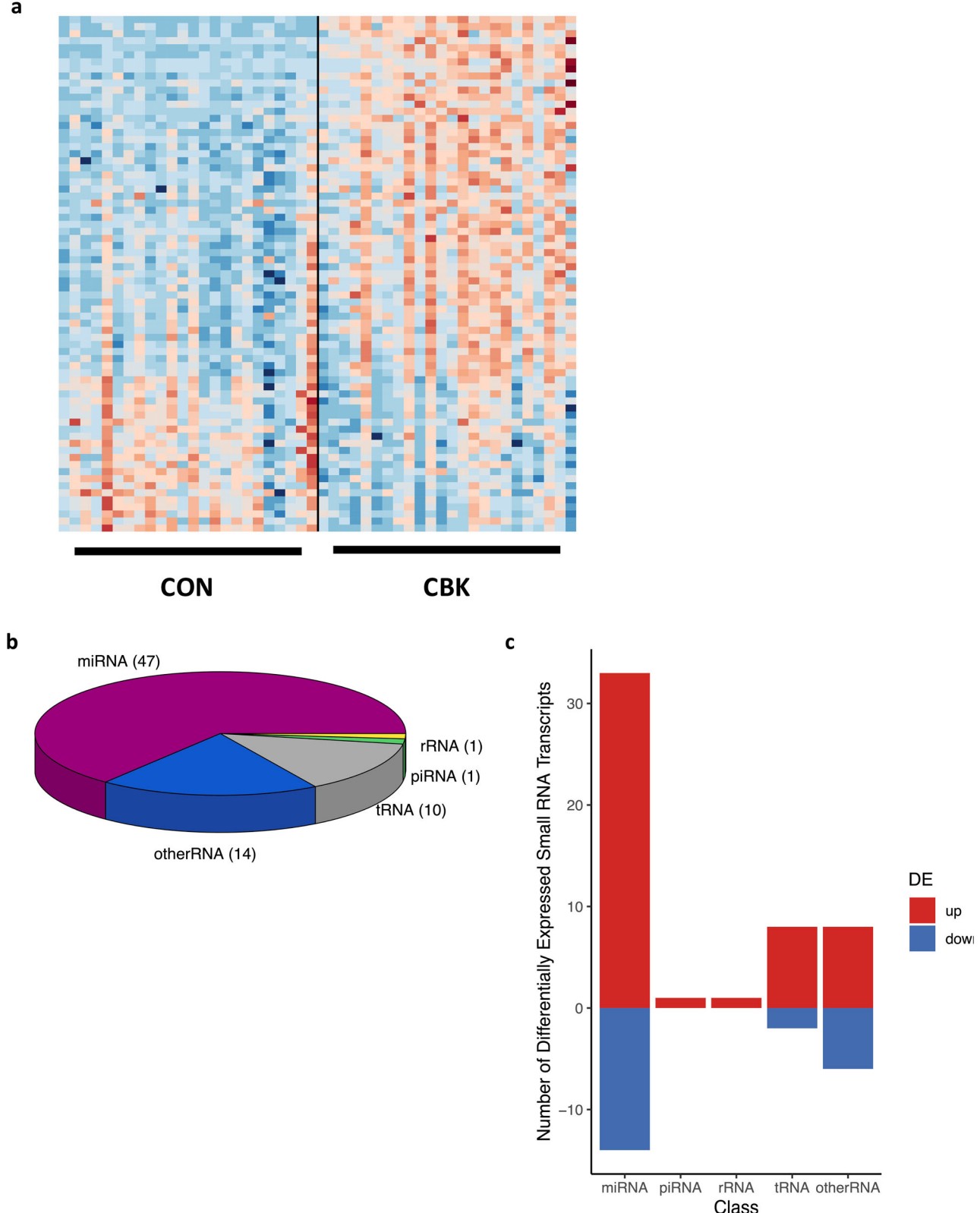

**Fig. 2 Alterations in small RNA species in hearts of CBK mice. a** Heat map for small RNA species exhibiting differential expression in CBK hearts (relative to CON hearts; $n = 24$). **b** Relative distribution of differentially expressed small RNA species classes in CBK hearts. **c** Number of induced (up) versus repressed (down) small RNA species that are differentially expressed (DE) in CBK (versus CON) hearts.

main effects of time-of-day for the 1929 small RNA species. In contrast, main effects of genotype were observed for 73 small RNA species (i.e., 3.78% of the small RNA species were differentially expressed in CBK versus CON hearts independent of

time-of-day; Fig. 2a and Supplemental Table 2). The small RNA species exhibiting a genotype main effect could be subclassified into miRNA (47), tRNA (10), rRNA (1), piRNA (1), and 'other' RNA (e.g., lncRNA, snoRNA, uncharacterized RNA; 14); see

Fig. 2b and Supplemental Table 2. It is noteworthy that 70% of these small RNA species were induced in CBK hearts, whereas 30% were repressed (Fig. 2c and Supplemental Table 2). Collectively, these data indicate that a discrete subset of small RNA species are differentially expressed in CBK hearts.

**Integrating the miRNome with the transcriptome of CBK hearts.** Given that miRNAs were among the most highly represented sub-classification of small RNA species that are differentially expressed in CBK hearts (Fig. 2b), we focused on miRNA species further. It has been estimated that miRNAs regulate up to two thirds of a cell's transcriptome[31]. Previously published studies suggest that the cardiomyocyte circadian clock influences 6–10% of the cardiac transcriptome[17,18]. We therefore hypothesized that one way in which the cardiomyocyte clock influences the cardiac transcriptome is through control of miRNAs. Here, we employed a bioinformatics approach to test this hypothesis. First, a sub-set of 21 differentially expressed miRNA species in CBK hearts was identified with a log2 fold change value of ±0.5, an adjusted *p*-value of less than 0.05, and average expression of greater than 50 counts (Fig. 3a and Table 1). Next, these 21 miRNAs were entered into miRNet (an online omics tool) to identify candidate target mRNAs; a total of 429 candidate mRNA species were identified. These mRNAs were subsequently compared to previously published RNA-seq data for CBK and littermate CON hearts[30]; criteria for differentially expressed mRNAs in CBK hearts included a log2 fold change value of ±0.5 and adjusted p-value of less than 0.05 (for mRNAs with an average expression of greater than 100 counts). Of the 573 differentially expressed mRNA species in CBK hearts (Supplemental Table 3), 268 (i.e., 46.8%) were identified as being putatively targeted by one or more of the 21 differentially expressed miRNAs in CBK hearts (Supplemental Table 4). A total of 377 unique interactions were identified between differentially expressed miRNA species that target differentially expressed mRNAs; Fig. 3b illustrates the inverse relationship for differential expression between the identified miRNAs and their candidate mRNA targets. Enrichr pathway analysis was next performed, revealing enrichment in pathways involved in circadian rhythms, cellular signaling, and metabolism (Fig. 3c). Collectively, this analysis raises the possibility that BMAL1 may indirectly influence the cardiac transcriptome in part through regulation of distinct miRNA species.

**Validation of miRNome and transcriptome data sets.** We sought to validate our omics observations through use of a second experimental methodology (RT-PCR) with an increased number of samples within each experimental group. With regards to the miRNome, 9 out of the 21 differentially expressed miRNAs included in the miRNome-transcriptome interaction analysis (Fig. 3 and Table 1) were selected for validation (based on those miRNAs with the highest number of mRNA interactions, and including only 1 miRNA species when 2 related isoforms were identified). RT-PCR confirmed that miR-1a-2–5p, miR-181a-1–3p, and miR-499-3p exhibit decreased levels in CBK hearts, relative to CON littermate hearts (i.e., genotype main effect; Fig. 4a). Similarly, increased levels of let-7c-1-3p, miR-23b-5p, miR-31-5p, miR-34a-5p, miR-215-5p, and miR-741-3p in CBK hearts were confirmed by RT-PCR (i.e., genotype main effect; Fig. 4a). Again, consistent with the RNAseq analysis, time-of-day main effects were not observed for the miRNA species interrogated by RT-PCR (with the exception of miR-215-5p; Fig. 4a). We next investigated the most highly differentially expressed mRNA targets of the 9 aforementioned miRNA species (Supplemental Table 4); these included *cdkn1a* (targeted by miR-1a-2-

5p), *npas2* (targeted by miR-181a-1-3p), *nt5e* (targeted by miR-499-3p), *armc2* (targeted by let-7c-1-3p), *herpud1* (targeted by miR-23b-5p), *mid1ip1* (targeted by miR-31-5p), *rhobtb1* (targeted by miR-34a-5p), *npc1* (targeted by miR-215-5p), and *mylk4* (targeted by miR-741-3p). Consistent with the transcriptomic data, RT-PCR revealed that *cdkn1a*, *nt5e*, and *npas2* mRNA levels are increased in CBK hearts, whereas *armc2*, *herpud1*, *mid1ip1*, *rhobtb1*, *npc1*, and *mylk4* mRNA levels are decreased (i.e., genotype main effects; Fig. 4b). Interestingly, the 2-way ANOVA also revealed main effects of time-of-day for *cdkn1a*, *npas2*, *herpud1*, *mid1lp1*, *rhobtb1*, and *mylk4* mRNA. Accordingly, cosinor analysis was performed, to assess whether 24 h oscillations differed between CBK and CON hearts for these 9 mRNA species. Supplemental Table 1 indicates that, with the exception of *n5te* and *npc1*, these mRNA species exhibit significant 24 h oscillations in CON hearts. In contrast, only *cdkn1a* and *rhobtb1* mRNA levels exhibit 24 h oscillations in CBK hearts; the amplitude of the 24 h oscillations for these two mRNAs were significantly attenuated in CBK hearts (relative to littermate CON hearts; Supplemental Table 1). Finally, we confirmed that protein levels of CDKN1A were significantly increased in CBK hearts at ZT0 (Fig. 4c). Collectively, these data validate our miRNome and transcriptome data sets, and suggest that alterations in miRNA species in CBK hearts are associated with attenuated/abolished 24 h rhythms in target mRNAs.

**Bioinformatic analysis identifies putative transcriptional regulators of miRNAs in CBK hearts.** To gain insight regarding possible mechanisms by which BMAL1 deletion leads to alterations in miRNA species in the heart, a bioinformatics approach was utilized. An initial interrogation of the differentially expressed transcripts in CBK hearts (Supplemental Table 3) failed to identify any mRNAs that encode for proteins with known functions in miRNA biogenesis/processing. As such, we focused on the possibility that distinct miRNAs were affected at a transcriptional level. We utilized the online omics tool Transmir v2.0 database to identify transcription factors (TF) that putatively regulate expression of the differentially expressed 21 miRNA species listed in (Table 1). We cross-referenced differentially expressed transcripts in CBK hearts (Supplemental Table 3) with the Transmir v2.0 database. This analysis identified a total of 32 unique TF-miRNA interactions, representing 8 unique TFs that putatively regulate 11 miRNAs; Fig. 5a is a visual representation of the differentially expressed TFs that potentially regulate a given miRNA species. For the 8 miRNAs that were repressed in CBK hearts, no common TFs were identified that readily explain miRNA differential expression; for example, the transcriptional repressor BHLHE40 exhibits decreased mRNA levels in CBK hearts, which would be anticipated to induce miRNA targets. In contrast, for 7 out of 13 miRNAs that were induced in CBK hearts, NR1D1 was identified as a candidate regulatory TF. Moreover, for 2 of these induced miRNA species, NR1D2 was similarly identified. Nr1d1 and Nr1d2 (encoding for REVERBα and REVERBβ, respectively) are established circadian clock components, that are directly regulated by BMAL1, and function as transcriptional repressors[32]. We have previously reported decreased *nr1d1* mRNA levels in these CBK hearts[29]; here, we similarly report decreased *nr1d2* mRNA levels in CBK hearts (genotype main effect; Fig. 5b). To investigate whether loss of REV-ERBα/β is sufficient to influence expression of the identified putative target miRNA species (independent of BMAL1), let-7c-1-3p, miR-23b-5p, miR-31-5p, miR-139-3p, miR-215-5p, miR-5123, and miR-7068-3p were assessed in cardiomyocyte-specific REV-ERBα/β double knockout (CM-RevDKO) and littermate control hearts collected at ZT10 (a time at which REV-ERBα/β

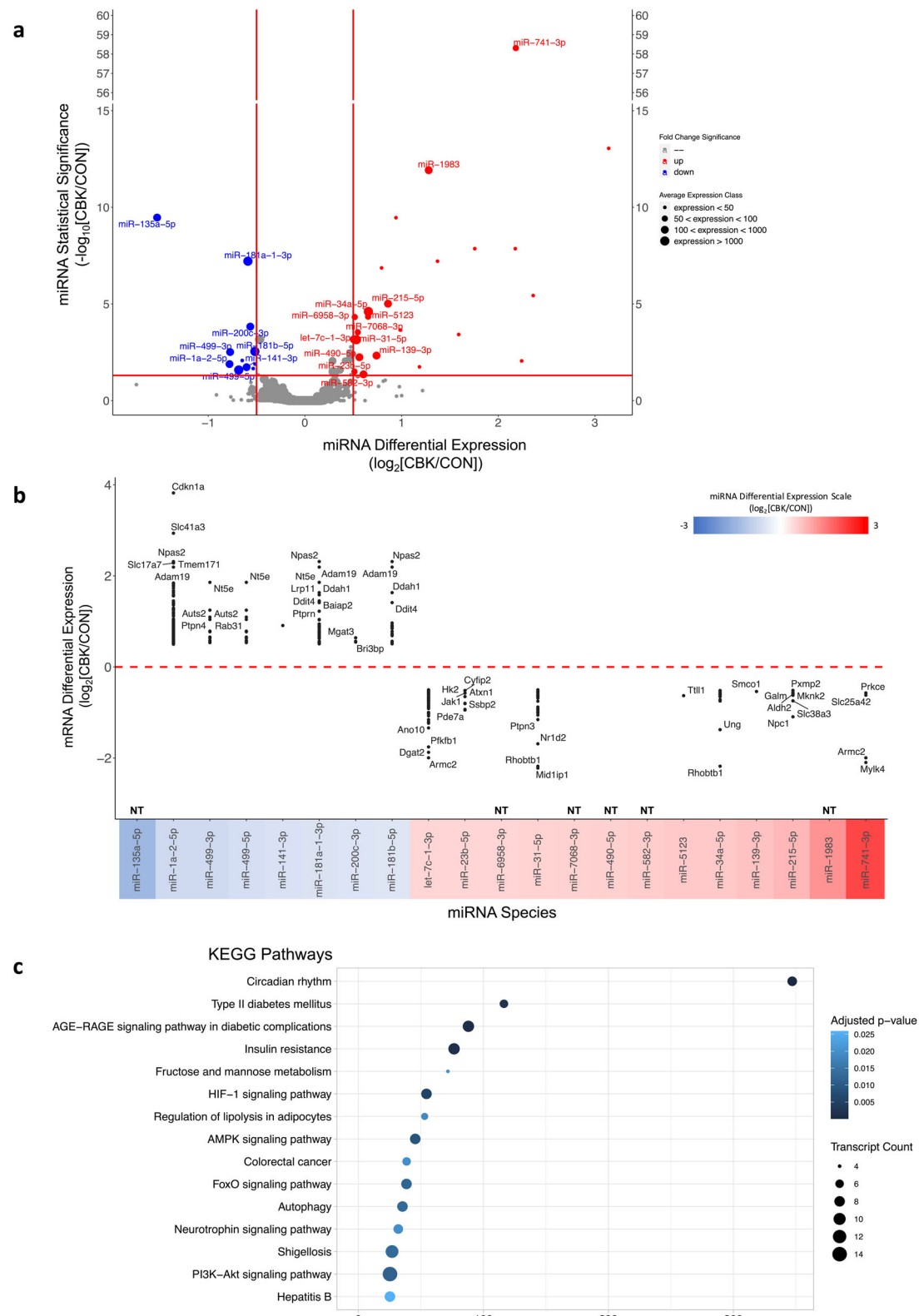

protein levels normally peak in the heart)[28]. Of these, let-7c-1-3p, miR-23b-5p, miR-139-3p, miR-5123, and miR-7068-3p exhibited increased levels in CM-RevDKO hearts (relative to CON hearts), whereas miR-31-5p and miR-215-5p were not differentially expressed (Fig. 5c). Next, comparison of the identified 54 putative mRNA targets for the 5 miRNA species that were induced in both CBK and CM-RevDKO hearts (Supplemental Table 4) with

recently published RNA-seq data for CM-RevDKO hearts[27] revealed that 19 transcripts were commonly repressed in both CBK and CM-RevDKO hearts (whereas none were induced; Fig. 5d). Of these 19 transcripts, 16 are putatively regulated by let-7c-1-3p (Fig. 5d and Supplemental Table 4). RT-PCR and cosinor analysis confirmed that 3 putative let-7c-1-3p targets (*dgat2*, *ptpn3*, and *usp54*) exhibit significant 24 h oscillations in CON at

**Fig. 3 Differential miRome-transcriptome interaction in CBK hearts. a** Volcano plot showing miRNA species with large and significant differences between CBK and littermate CON hearts. The x-axis shows Log2 of fold change (CBK/CON), while the y-axis shows Log2 of adjusted p-values for differential expression. Size of circle represents level of expression (i.e., average counts), while color represents miRNAs that are induced (up; red) versus repressed (down; blue) in CBK hearts. Horizontal line represents an adjusted p-value of 0.05, while the vertical lines represent a Log2 fold change of ±0.5. **b** Differential expression of predicted mRNA targets of the differentially expressed miRNA species in CBK hearts. The x-axis shows the differentially expressed miRNAs, while the y-axis shows Log2 of fold change (CBK/CON) for the target mRNA species. Color represents level of miRNA differential expression (based on the scale bar). **c** Pathway analysis of differentially expressed mRNAs that interact with differentially expressed miRNA species; the x-axis represents the Enrichr Combined Enrichment Score, which accounts for the number of differentially expressed transcripts in CBK hearts compared to the total number of transcripts in the respective KEGG Pathway.

| Table 1 Differentially expressed miRNA species in CBK hearts. | | | | |
|---|---|---|---|---|
| **miRNA Species** | **Control Expression** | **CBK Expression** | **Fold Change** | **Adjusted p-value** |
| let-7c-1-3p | 312.38 ± 17.71 | 443.60 ± 23.19 | 1.420 | 6.79E-04 |
| miR-135a-5p | 1,408.40 ± 122.52 | 488.01 ± 51.89 | 0.347 | 3.40E-10 |
| miR-139-3p | 101.85 ± 9.13 | 170.26 ± 17.77 | 1.672 | 4.67E-03 |
| miR-141-3p | 264.54 ± 24.50 | 174.18 ± 13.40 | 0.658 | 1.87E-02 |
| miR-181a-1-3p | 1,567.57 ± 54.59 | 1,042.48 ± 58.36 | 0.665 | 6.12E-08 |
| miR-181b-5p | 3,533.32 ± 248.30 | 2,468.51 ± 156.89 | 0.699 | 2.87E-03 |
| miR-1983 | 146.60 ± 14.20 | 355.24 ± 29.68 | 2.423 | 1.20E-12 |
| miR-1a-2-5p | 189.88 ± 19.56 | 110.67 ± 11.57 | 0.583 | 1.29E-02 |
| miR-200c-3p | 208.34 ± 11.65 | 141.26 ± 8.55 | 0.678 | 1.49E-04 |
| miR-215-5p | 207.27 ± 16.65 | 375.70 ± 25.37 | 1.813 | 9.70E-06 |
| miR-23b-5p | 41.48 ± 3.56 | 59.36 ± 4.19 | 1.431 | 3.16E-02 |
| miR-31-5p | 1,696.59 ± 82.65 | 2,455.65 ± 160.85 | 1.447 | 7.03E-04 |
| miR-34a-5p | 12,684.10 ± 787.89 | 19,995.12 ± 1,147.87 | 1.576 | 2.46E-05 |
| miR-490-5p | 80.95 ± 5.62 | 120.00 ± 8.13 | 1.482 | 5.71E-03 |
| miR-499-3p | 613.57 ± 56.67 | 358.85 ± 33.73 | 0.585 | 3.08E-03 |
| miR-499-5p | 37,782.93 ± 3,290.69 | 23,508.22 ± 2,253.45 | 0.622 | 2.61E-02 |
| miR-5123 | 55.14 ± 4.14 | 86.21 ± 4.25 | 1.563 | 4.55E-05 |
| miR-582-3p | 122.19 ± 17.83 | 185.81 ± 13.94 | 1.521 | 4.37E-02 |
| miR-6958-3p | 45.05 ± 2.33 | 64.58 ± 2.65 | 1.433 | 4.75E-05 |
| miR-7068-3p | 77.42 ± 3.83 | 113.00 ± 7.52 | 1.460 | 2.90E-04 |
| miR-741-3p | 23.50 ± 1.70 | 106.00 ± 5.37 | 4.511 | 4.91E-59 |

Hearts were isolated from CBK and littermate CON hearts at 4 h intervals over a 24 h period, followed by small RNAseq. Two-way ANOVAs were performed using DESeq2A, to identify differentially expressed miRNA species between CBK and CON hearts. A sub-set of differentially expressed miRNA species in CBK hearts was subsequently identified with a log2 fold change value of ±0.5, an adjusted p-value of less than 0.05, and an average expression of greater than 50 counts; these miRNA species are presented in the table. All data are presented as mean ± SEM.

the mRNA level; these oscillations were either abolished (*dgat2*, *ptpn3*) or significantly altered (*usp54*) in CBK, hearts (Fig. 5e and Supplemental Table 1). Moreover, *dgat2*, *ptpn3*, and *usp54* mRNA levels were decreased in CBK hearts independent of the time-of-day (i.e., genotype main effect; Fig. 5e). RT-PCR similarly confirmed decreased expression of *dgat2*, *ptpn3*, and *usp54* mRNA levels in CM-RevDKO hearts at ZT10 (relative to littermate CON hearts; Fig. 5f). Finally, to investigate the potential off-target contribution of Cre in cardiomyocytes, hearts were collected from MHCα-Cre positive and littermate CON mice at ZT10; RT-PCR analysis revealed no differences in let-7c-1-3p miRNA, *ptpn3* mRNA, and *usp54* mRNA levels, between MHCα-Cre and CON mice (Fig. 5g). In contrast, a slight (18.5%) decrease in *dgat2* mRNA levels were observed in MHCα-Cre hearts (Fig. 5g). Collectively, these data suggest that the BMAL1–REV-ERBα/β axis potentially regulates a subset of transcripts (such as *dgat2*, *ptpn3*, and *usp54*) through distinct miRNA species (such as let-7c-1-3p).

## Discussion
The purpose of the present study was to interrogate perturbations in the cardiac miRNome following genetic disruption of the cardiomyocyte circadian clock (through deletion of Bmal1). Here, we report differential expression of 47 miRNA species in cardiomyocyte-specific Bmal1 knockout (i.e., CBK) hearts

(relative to CON hearts), of which 70% are induced. When applying more stringent criteria to identify miRNA species with potentially greater biologic impact (i.e., higher level of expression in CON heart, as well as higher level of differential expression in CBK hearts), 21 candidate miRNAs emerged; 8 were repressed, while 13 were induced in CBK hearts. Bioinformatic analysis predicted that these 21 miRNA species influence diverse biological processes involved in circadian rhythms, intracellular signaling, and metabolism. Subsequent analyses highlighted the circadian clock components REV-ERBα/β as putative transcriptional modulators of 38.5% of the miRNA species that were induced in CBK hearts (i.e., let-7c-1-3p, miR-23b-5p, miR-139-3p, miR-5123, and miR-7068-3p); we speculate that disruption of this putative BMAL1–REV-ERBα/β–miRNA regulatory network may contribute towards repression of a subset of cardiac transcripts following ablation of the cardiomyocyte circadian clock.

The cardiomyocyte circadian clock (CCC) is essential for maintenance of normal cardiac function[33]. Animal- and cell-based studies have highlighted that this cell autonomous mechanism temporally governs mRNA and protein levels of numerous mediators/regulators of cardiac signaling, metabolism, electrophysiology, and contractility[17,18,27,34–41]. Examples include p85α (regulatory subunit of PI3K, a kinase central to multiple signaling cascades), DGAT2 (diacylglycerol acyltransferase 2, involved in triglyceride synthesis), NAMPT (nicotinamide phosphoribosyltransferase, a component of the NAD salvage

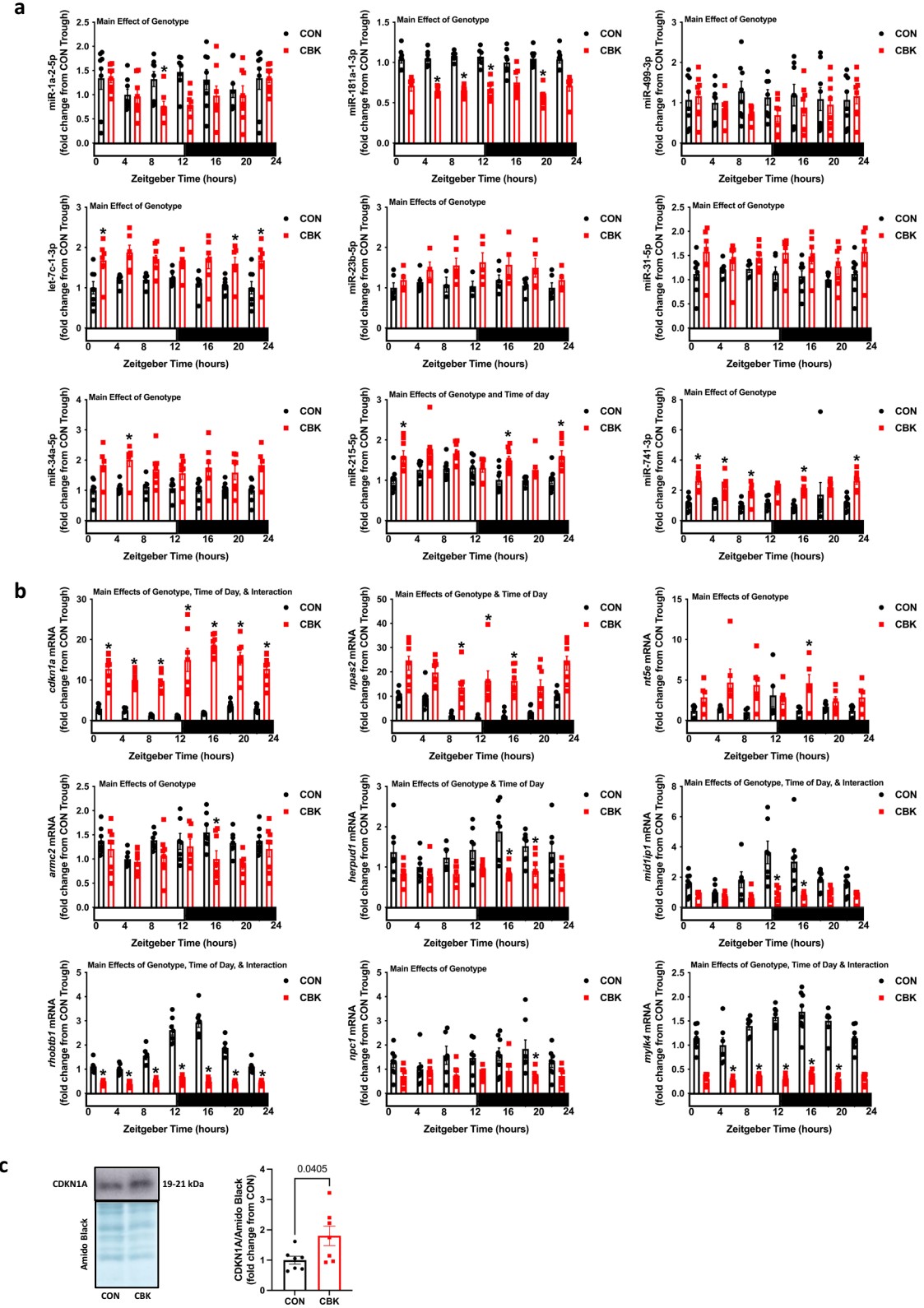

**Fig. 4 Differential expression of candidate miRNAs and their target mRNAs in CBK hearts. a** Diurnal variations in miR-1a-2-5p, miR-181a-1-3p, miR-499-3p, let-7c-1-3p, miR-23b-5p, miR-31-5p, miR-34a-5p, miR-215-5p, and miR-741-3p levels for CBK and CON hearts collected at six distinct times of the day (n = 3–8). **b** Diurnal variations in *cdkn1a, npas2, nte5, armc2, herpud1, mid1ip1, rhobtb1, npc1,* and *mylk4* mRNA levels for CBK and CON hearts collected at six distinct times of the day (n = 4–8). **c** Western blot of p21 protein levels in CBK and littermate control (CON) hearts collected from 12wk old mice at ZT0 (n = 7). Data are presented as mean ± SEM. Main effects of genotype, time-of-day, and/or interaction are reported at the top of the figure panels. *p*-values are either reported as numbers, or as symbols: *p < 0.05 for CBK versus CON hearts.

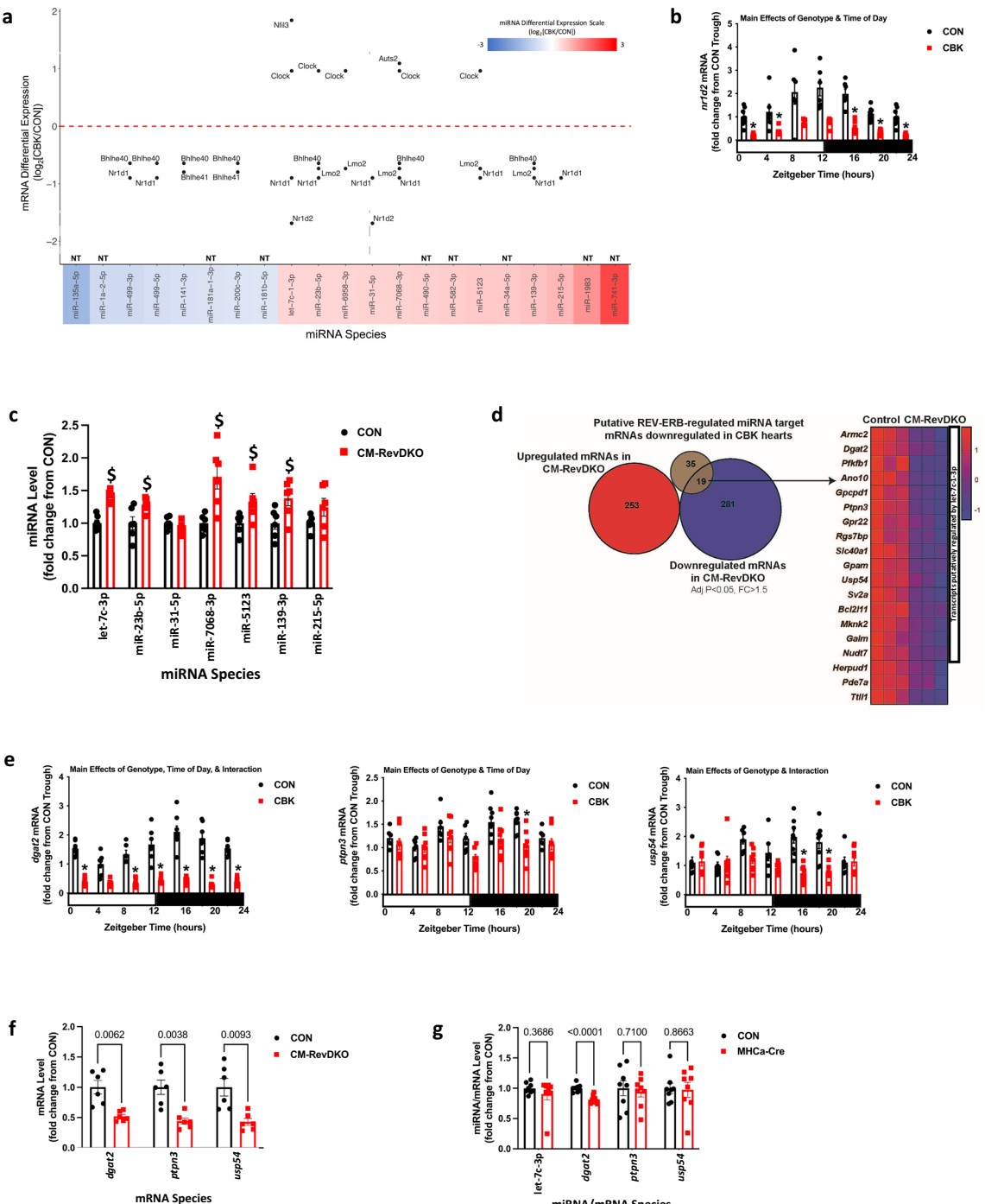

**Fig. 5 Identification of REVERBα/β as a potential regulator of miRNA species in the heart. a** Differential expression of predicted transcription factors regulating differentially expressed miRNA species in CBK hearts. The x-axis shows the differentially expressed miRNAs, while the y-axis shows Log2 of fold change (CBK/CON) for the predicted transcription factors. Color represents level of miRNA differential expression (based on the scale bar). **b** Diurnal variations in nr1d2 mRNA levels for CBK and CON hearts collected at six distinct times of the day (n = 6–8). **c** RT-PCR for let-7c-1-3p, miR-23b-5p, miR-31-5p, miR-139-3p, miR-215-5p, miR-5123, and miR-7068-3p levels in CM-RevDKO and littermate control (CON) hearts collected at ZT10 (n = 6).
**d** Bioinformatic analysis identified 19 transcripts that are repressed in both CBK and CM-RevDKO hearts, which are predicted to be regulated by let-7c-1-3p, miR-23b-5p, miR-139-3p, miR-5123, and miR-7068-3p; the first 16 transcripts are putative let-7c-1-3p targets. **e** Diurnal variations in dgat2, ptpn3, and usp54 mRNA levels for CBK and CON hearts collected at six distinct times of the day (n = 6–8). **f** RT-PCR for dgat2, ptpn3, and usp54 mRNA levels in CM-RevDKO and littermate control (CON) hearts collected at ZT10 (n = 6). **g** RT-PCR for let-7c-1-3p miRNA, as well as dgat2, ptpn3, and usp54 mRNA, levels in CM-RevDKO and littermate control (CON) hearts collected at ZT10 (n = 8). Data are presented as mean ± SEM. Main effects of genotype, time-of-day, and/or interaction are reported at the top of the figure panels. p-values are either reported as numbers, or as symbols: *p < 0.05 for CBK versus CON hearts; $p < 0.05 for CM-RevDKO versus CON hearts.

pathway), HCN4 (hyperpolarization activated cyclic nucleotide gated potassium channel 4, involved in setting heart rate), SCN5a (sodium voltage-gated channel alpha subunit 5, central ion channel involved in the action potential), and TCAP (titin-cap, a sarcomeric protein involved in contractility)[18,27,35,39,40]. Consistent with this, 24 h oscillations in cardiac insulin signaling, lipid metabolism, glucose oxidation, heart rate, and contractile function, are invariably altered following manipulation of the CCC[17,36–38,41]. Although progress has been made, the precise mechanistic links between the CCC and the aforementioned cardiac processes remain incompletely defined. In some instances, evidence exists in support of the concept that circadian clock components directly regulate target genes encoding for mediators/regulators of cardiac function. For example, the BMAL1/CLOCK heterodimer directly activates transcription of p85a, Nampt, and Hcn4 genes[18,40,42]. However, the possibility exists that the circadian clock employs additional mechanisms for temporally controlling cellular processes. One such mechanism potentially includes miRNAs. Recent studies in the extracardiac tissues highlight an interconnected relation between miRNA species and the circadian clock. More specifically, distinct miRNA species are directly regulated by the circadian clock, which feedback onto specific circadian clock components[21,22]. It has been suggested that clock control of miRNAs fine tune the phase and/or amplitude of rhythmic mRNA targets (and in some cases, may even induce oscillations in protein levels from non-rhythmic mRNA species)[21–24]. Despite appreciation that miRNAs impact numerous cellular functions in the heart[25], it is currently unknown whether miRNAs serve as a mechanistic link between the CCC and cardiac physiology.

Mimicking alterations in 24 h environmental (e.g., light/dark cycles) and behavioral (e.g., fasting/feeding cycles) factors that shift workers might be exposed to perturbs circadian clock gene oscillations in the heart, in association with adverse cardiac remodeling[26,43–46]. Similarly, selective disruption of the CCC through genetic means results in adverse cardiac remodeling, heart failure, and shortened lifespan[17,18,27]. What remains less clear are the mechanisms by which circadian disruption leads to cardiac disease. To gain insight regarding the possible contribution of miRNA species, an unbiased omics-based approach was pursued initially. Using CBK mice as a model of CCC disruption (Fig. 1), small RNA-seq revealed that surprisingly few small RNA species (<4%) were significantly impacted following impairment of this molecular timekeeping mechanism. Of these, miRNAs were the primary sub-classification of small RNA species that were differentially expressed in CBK hearts, accounting for 47 (out of 73) species with significantly altered levels (Fig. 2b). Of the 47 differentially expressed miRNA species, the majority (70%) were induced (Fig. 2c). Similarly, when more stringent criteria were applied in an attempt to identify biologically-relevant perturbation in miRNA species, 13 out of 21 differentially expressed miRNA species were induced in CBK hearts (i.e., 62%; Fig. 3a). Given that binding of BMAL1 to promoter regions typically enhances transcription, the induction of miRNA species following BMAL1 deletion was considered likely to be through an indirect mechanism. Consistent with this idea, bioinformatic analysis for prediction of transcriptional modulators of differentially expressed miRNAs identified REV-ERBα/β as a primary candidate (Fig. 5a). Given that the genes encoding for REV-ERBα/β (i.e., Nr1d1 and Nr1d2) are direct targets of the BMAL1/CLOCK heterodimer[32], that REV-ERBα/β levels are decreased in CBK hearts at both mRNA and protein levels (Fig. 5b)[18,28,29], and that REV-ERBα/β typically serve as repressors, we decided to interrogate these transcriptional modulators further. Previous studies report that cardiomyocyte-specific deletion of both REV-ERBα and REV-ERBβ results in disruption of the circadian clock[27].

Using this model, we observed that similar to CBK hearts, CM-RevDKO hearts exhibit increased expression of let-7c-1-3p, miR-23b-5p, miR-139-3p, miR-5123, and miR-7068-3p (Fig. 5c). Since REV-ERBα/β levels are reduced in both the CBK and CM-RevDKO hearts, whereas BMAL1 levels are highly repressed and induced in CBK and CM-RevDKO hearts, respectively[18,27], the data are most consistent with a model in which REV-ERBs directly repress these miRNAs, while BMAL1 has the critical but indirect role of driving circadian expression of the REV-ERBs.

It is noteworthy that 19 of the predicted mRNA targets for the aforementioned 5 miRNA species were repressed in both CBK and CM-RevDKO hearts (Fig. 5d); these mRNAs have established functions in metabolism (e.g., dgat2), ion transport (e.g., ano10), and signaling (e.g., ptpn3, usp54)[47–49]. Given the fundamental importance of these cellular functions, it is tempting to hypothesize that induction of these miRNAs contributes towards the age-onset adverse cardiac remodeling and heart failure observed in both CBK and CM-RevDKO mice. Indeed, elevated miR-23b levels have previously been reported to induce cardiac hypertrophy, a phenotype observed in both CBK and CM-RevDKO hearts[18,27,30,50,51]. Interestingly, of the 19 putative miRNA targets that were induced both CBK and CM-RevDKO hearts, 16 are known to be regulated by let-7c-1-3p, thus highlighting the BMAL1–REV-ERBα/β–let-7c-1-3p relationship as a potential mechanism by which the circadian clock regulates cardiac processes.

To the best of our knowledge, the current study is the first to investigate regulation of miRNA species by the CCC. However, several noteworthy limitations exist. This includes the descriptive nature of the experimental design, which identifies associations, as opposed to establishment of mechanistic links. Indeed, our major findings include the observations that deletion of BMAL1 and/or REV-ERBα/β lead to increased expression of 5 common miRNA species, which is associated with decreased expression of putative target mRNA species. Future studies are required to determine the extent to which alterations in mRNA levels in CBK and/or CM-RevDKO hearts are caused by changes in miRNA species. Similarly, the current study is unable to confirm that increased expression of the 5 miRNA species described are due directly to decreased levels of REV-ERBα/β (or through an, as yet, unidentified mechanism). It is also noteworthy that miRNAs impact cell function not only through alterations in mRNA stability, but also by influencing translation[20]. The possibility therefore exists that the importance of the circadian BMAL1–REV-ERBα/β regulatory network on miRNAs studied here has been underestimated by interrogating only the transcriptome. Future studies are required to investigate the contribution of clock control of miRNA species on the cardiac proteome. The current study has also not directly investigated the functional consequences of altered miRNA levels in the heart (e.g., does chronic induction/repression of miRNA species in CBK and/or CM-RevDKO hearts contribute towards age-onset adverse cardiac remodeling and premature death reported in these models?), nor has it investigated whether similar alterations in cardiac miRNA species are observed in non-genetic models of circadian disruption (e.g., shift work models). Finally, whether circadian control of miRNA species occurs in either a sex or cell type specific manner remains unknown.

In summary, the current omics-based study highlights that cardiomyocyte-specific disruption of BMAL1 results in differential expression of cardiac miRNAs that are predicted to influence circadian rhythms, cellular signaling, and metabolism. Moreover, bioinformatic and experimental findings suggest that a subset of the induced miRNAs following BMAL1 deletion may be secondary to attenuated REV-ERBα/β. We speculate that induction of let-7c-1-3p, miR-23b-5p, miR-139-3p, miR-5123, and

miR-7068-3p in the heart following genetic ablation of BMAL1 or REV-ERBα/β may contribute towards the development of age-onset cardiomyopathy.

## Methods

**Animal models**. The present study utilized cardiomyocyte-specific Bmal1 knockout (CBK; Bmal1$^{flox/flox}$/MHCαCre$^{+/−}$) and littermate control (CON; Bmal1$^{flox/flox}$/MHCαCre$^{−/−}$) mice, in addition to cardiomyocyte-specific Rev-erbα/β double knockout (CM-RevDKO; Rev-erbα$^{flox/flox}$/Rev-erbβ$^{flox/flox}$/MHCαCre$^{+/−}$) and littermate control (CON; Rev-erbα$^{flox/flox}$/Rev-erbβ$^{flox/flox}$/MHCαCre$^{−/−}$) mice. Cardiomyocyte-specific Cre positive (MHCα-Cre; MHCαCre$^{+/−}$) and littermate wild-type control (CON; MHCαCre$^{−/−}$) mice were also utilized. These mouse models have been described previously[18,27]. All experimental mice were 12wk old male mice, and were housed under temperature-, humidity-, and light- controlled conditions, at either the Center for Comparative Medicine at the University of Alabama at Birmingham (USA) or Max Planck Institute for Heart and Lung Research (Germany). A 12 h light/12 h dark cycle was enforced (lights on at zeitgeber time [ZT] 0); the light/dark cycle was maintained throughout these studies, facilitating investigation of diurnal variations. Mice were housed in standard microisolator cages and received food and water ad libitum. All animal experiments were approved by the Institutional Animal Care and Use Committee of the University of Alabama at Birmingham and by the responsible Committee for Animal Rights Protection of the State of Hessen (Regierungspraesidium Darmstadt, Wilhelminenstr. 1–3, 64283 Darmstadt, Germany) with project number B2-2018. We have complied with all relevant ethical regulations for animal testing.

**RNA isolation and quantitative RT-PCR**. Two RNA preparations were generated from biventricular samples. First, total RNA was isolated using previously described standard procedures[52]; this RNA was utilized for assessment of mRNA species. Second, small RNA species enriched preparations were generated from biventricular samples using the miRNeasy kit (Qiagen); this RNA was utilized for assessment of small RNA species (e.g., snoRNAs and miRNAs). Assessment of candidate mRNA and small RNA species was performed by quantitative RT-PCR, using previously described methods[53,54]. For mRNA quantitative RT-PCR, either commercially available (armc2, cdkn1a, herpud1, hlf, mid1ip1, mylk4, npc1, nr1d2, nt5e, ptpn3, rhobtb1, tef, usp54; ThermoFisher Scientific) or custom-designed (actb, cry1, cry2, dgat2, npas2, per1, per3, ppia, rplp0) TaqMan assays were utilized; primer and probe sequences for custom-designed TaqMan assays have been reported previously[29,55]. For miRNA and snoRNA quantitative RT-PCR, commercially available (let-7c-1-3p, miR-1a-2-5p, miR-139-3p, miR-181a-1-3p, miR-215-5p, miR-23b-5p, miR-31-5p, miR-34a-5p, miR-499-3p, miR-5123, miR-741-3p, miR-7068-3p, sno202, sno234) TaqMan assays were utilized (ThermoFisher Scientific). Data for mRNA species were normalized to the combined expression of three housekeeping mRNAs (actb, ppia, rplp0); data for miRNA species were normalized to the combined expression of two snoRNAs (sno202, sno234). All quantitative RT-PCR data are presented as fold change from an indicated control group.

**Western blotting**. Qualitative analysis of protein expression was performed via standard western blotting procedures, as described previously[37]. Briefly, 15–30 µg protein lysate was separated on polyacrylamide gels and transferred to PVDF membranes. Membranes were probed for with anti- BMAL1 and CDKN1A antibodies (Abcam ab3350 and Santa Cruz sc-6246, respectively). Rabbit and mouse HRP-conjugated secondary antibodies (Cell Signaling 7074 and 7076, respectively) were used for chemiluminescent detection

with Luminata Forte Western Blotting substrate (Millipore, WBLUF0100). All densitometry data were normalized to amido black staining. Importantly, in order to minimize the contribution that position on the gel might have on outcomes, samples were randomized on gels; samples were re-ordered post-imaging, only for the sake of illustration of representative images (note, all bands for representative images for an individual experiment were from the same gel; original images are presented in Supplemental Fig. 1).

**Small RNA sequencing**. Unbiased screening of small RNA species was performed through small RNA sequencing (small RNA-seq) by the UAB Genomics Core facility; this core utilized the Qiagen miRNA sequencing platform. In short, small RNA enriched total RNA was extracted and small RNA sequencing libraries were prepared using the QIAseq miRNA library kit. Universal cDNA synthesis was next performed, followed by cDNA cleanup, library amplification, and library cleanup. After sequencing, data were uploaded to the GeneGlobe data analysis center (proprietary Qiagen website), which mapped and counted the unique molecular identifiers using miRBase.

**Sequencing data analysis and bioinformatics**. Small RNA-seq data analysis was performed on transcripts that had an average (for all heart samples, independent of experimental group) expression value > 10 counts. Variance was calculated for each individual sample after scaling all transcript targets by dividing each value by the standard deviation; no samples were defined as outliers as all sample variances were within two standard deviations from the mean variance for a given experimental group. Two-way ANOVAs were performed on small RNA-seq data using DESeq2[56]. The miRNet online omics tool was utilized to generate a list of mRNA targets putatively regulated by candidate miRNAs identified; this miRNet tool utilized miRNA target gene data from multiple annotated databases (miRTarBase v8.0, TarBase v8.0, and miRecords)[57]. The online omics tool Transmir v2.0 database was utilized to identify transcription factors (TF) that putatively regulate the expression of differentially expressed miRNA species[58]. For both miRNet and Transmir, the distinction between −5p and 3p for a given miRNA was not considered during database cross referencing. Enrichr was utilized for pathway analysis of candidate mRNAs that were potentially regulated by miRNAs[59]. Several R packages were utilized to visualize the aforementioned analyses, including: (1) the "RColorBrewer" package for generating heatmaps within base R; (2) "ggplot2", "ggrepel", and "ggbreak" packages for generating Volcano plots; and (3) the "igraph" package for generating network interaction plots. The online tools miRBase and the UCSC Genome Browser were also utilized to confirm the position within the genome that encodes for miRNA species of interest.

**Statistics and reproducibility**. Statistical analyses of non-transcriptomic data (i.e., not RNA-seq data) were performed using Prism statistical software. Briefly, main effects of genotype (i.e., CBK versus CON) and time (i.e., ZT) were determined using one-way and two-way ANOVAs (where applicable). Normality of data was assessed through use of the Shapiro–Wilks test, followed by either parametric (t-tests for only two experimental groups or Sidak's post-hoc test for multiple pairwise comparisons) or non-parametric (Mann–Whitney for only two experimental groups or Kruskal–Wallis with Dunn's correction for multiple pairwise comparisons). Cosinor analyses were performed to determine whether 24 h time series data significantly fit a cosine curve; if they did, then mesor (daily average value), amplitude (peak-to-mesor difference), and acrophase (timing of the peak) were calculated and compared between experimental groups, as described

previously[44]. In all analyses, the null hypothesis of no model effects was rejected at $p < 0.05$.

**Reporting summary**. Further information on research design is available in the Nature Portfolio Reporting Summary linked to this article.

## Data availability

Numerical source data underlying graphs and plots in the manuscript can be found on supplementary data file. All sequencing data can be accessed in GEO, using accession number GSE237168.

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

## Acknowledgements

We would like to thank Abigail Constant and Yannick N. Mangold for technical support. This work was supported by the National Heart, Lung, and Blood Institute (HL149159, HL154531 and HL007081) and the National Institute of Diabetes and Digestive and Kidney Diseases (DK45586).

## Author contributions

M.N.L., M.A.L., P.D., and M.E.Y. designed research. M.N.L., L.J.W., G.S., B.J.C., and P.D. performed research. M.N.L., L.J.W., P.D., and M.E.Y. analyzed data. L.J.W. and M.E.Y. drafted the manuscript. M.N.L., L.J.W., M.A.L., P.D., and M.E.Y. edited the manuscript.

## Competing interests

The authors declare no competing interests.
