## [Peer review file · Communications Biology]

Cardiomyocyte-Specific Disruption of the Circadian BMAL1– REV-ERB α/β Regulatory Network Impacts Distinct miRNA Species in the Murine HeartReviewers' comments:

Reviewer #1 (Remarks to the Author):

“Cardiomyocyte-Specific Disruption of the Circadian BMAL1–REV-ERBa/b Regulatory Network Impacts Distinct miRNA Species in the Murine Heart” by Latimer, Williams, and colleagues works to identify the role that the cardiomyocyte circadian clock (CCC) disruption impacts small RNAs. The group utilizes cardiomyocyte-specific Bmal1 knockout (CBK) mice to identify differentially expressed miRNAs in the CBK mouse. Several of the miRNAs are predicted to be regulated by the REV-ERBa/b. The cardiomyocyte-specific Rev-ervalpha/beta double knockout mouse (CM-RevDKO) also showed increased expression in several of the miRNAs as the CBK mouse. The authors conclude disruption of the BMAL1-REV-ERValpha/beta regulatory network can modify miRNAs that influence cell function. This work provides new information beyond the transcriptomic landscape and is expected to significantly impact the field. It opens avenues for future research that facilitate the identification of new mechanisms for circadian regulation of mRNA transcripts, translation, and protein levels.

The authors do a good job listing the limitation in the Discussion section. However, several additional limitations should be included.

The study focuses only on male mice, and it remains unclear if this regulatory network exists in female mice.

Although the authors use cardiomyocyte-specific transgenic models, the tissue analysis contains many different cell types. Depending on the cell type-specific or level of expression for different RNAs, some cardiomyocyte-specific differences in RNA expression might not be resolved. Cell type differences in levels of RNA expression could help explain why only so few small RNA species seemed to be impacted in heart tissue from the transgenic mice. The absolute number of RNA impacted in cardiomyocytes may be much higher.

Minor

Although most of the miRNAs explored in the CM-RevDKO mouse do not show a significant time-of-day interaction in the CON or CBK mice, why only test ZT 10 in the CM-RevDKO mice? Especially since there is the time of day that mire-215-5p is likely not expected to be different at this time based on data in Figure 4.

Figure 5 Bvi panel has text hidden behind the graph.

In the figure legends, “. x-axis shows” Could be “The x-axis...” if one wants to avoid capitalizing the X.” Also, “eight distinct time of the day” could be stated more clearly or make “time” plural.

References are a different font than the main manuscript.

Reviewer #2 (Remarks to the Author):

In this manuscript, Latimer and colleagues aim to assess the extent to which the circadian clock impacts the expression of micro-RNAs in the heart. This study is set in the context that while cardiomyocyte circadian clock (CCC) disruption is known to cause cardiac pathologies, the mechanisms remain poorly understood. The authors postulate that because miRNAs can act as critical modulators of cardiac physiology/pathology they may play a role in the development of CCC disruption disorders. To assess this, the authors performed unbiased small RNA screens in 12-week-old wildtype and cardiomyocyte-specific Bmal1 knockout mice. This revealed a several differentially expressed micro-RNA's, including a number that were under the transcriptional control of REV-ERBa/b. They then used REV-ERBa/b double KO mice to confirm its regulatory control of these small RNA's. Finally, they highlight that one miRNA, let-7c-1-3p, is responsible for the regulation of several genes that are differentially expressed in the CCC disrupted heart. As a result of these observations the authors suggest the existence of a Bmal1/REV-ERBa/b pathway that regulates miRNA expression prior to the onset of CCC disruption -induced pathology in the heart. Overall, this is a largely descriptive study but it is important as the study is the first of its kind to link circadian disruption to micro-RNA dysregulation in the heart and lays the groundwork for future investigations to help clarify this mechanistically.

Major Comments:

1. As mentioned by the authors, the cardiomyocyte specific Bmal1 KO mice have been previously characterized. It is not clear why in figure 1 they have chosen to add the analysis of additional genes that do not seem to appear anywhere else in the manuscript. In a way, it makes the inclusion of this data unnecessary. Please expand upon why these genes are important or consider removing the data.

2. The authors conclude from the data presented in Figure 2 that the main effect on miRNA expression is not time of day dependent, but rather genotype dependent. Importantly though, in both animal models used by the authors they compare KO (cre+/-) to WT (cre-/-) animals. Knowing that Cre-recombinase induces several unrelated effects in the cardiomyocyte, could the authors provide data about Cre-only effects on miRNA expression.

3. For Figure 5, including an analysis of the mRNA expression in the CBK mice of the let-7c-1-3p targets would greatly help in demonstrating the importance of this miRNA.
4. Any mechanistic insights (i.e. mir-mimic's in mice or culture) demonstrating that at least one of the miRNA's like let-7c-1-3p regulate the expression of the particular genes and that this results in some sort of pathology would greatly increase the impact of this work.
5. Alternatively, given that the authors chose to study what was happening prior to the onset of any type of CCC disruption-induced pathology, a screen of these miRNA's that demonstrates that they are also enriched in the CBK or REV-ERBa/b DKO diseased hearts would be helpful in demonstrating their importance.

Minor comments:

1. Can the authors please comment on how the miRNA's in Figure 2 only show a genotype-dependent difference, while their gene targets in Figure 3 show both a time of day and genotype effect?
2. Blots provided in supplemental data appear to have a lot of non-specific binding for Bmal1, with the lowest intensity band being indicated as Bmal1. Please provide a positive control.
3. The representative blots shown in Figure 1Ai and 3 Ci should be improved.

Response to Editor's and Reviewer's Comments

Title: Cardiomyocyte-Specific Disruption of the Circadian BMAL1–REV-ERB α/β Regulatory Network Impacts Distinct miRNA Species in the Murine Heart

Authors: Mary N. Latimer, Lamario J. Williams, Gobinath Shanmugan, Bryce J. Carpenter, Mitchell A. Lazar, Pieterjan Dierickx, Martin E. Young

Manuscript ID: COMMSBIO-23-2474

General Comments to the Editor and Reviewers

The authors are grateful to the Editor and the Reviewers for their constructive comments. The manuscript has been revised in response to these comments, as outlined in detail below. This includes the addition of new data within Figure 5, indicating: 1) the extent to which three putative let-7c-1-3p target mRNAs are altered in CBK and CM-RevDKO hearts (relative to littermate controls); and 2) that Cre alone is not responsible for alterations in cardiac levels of either let-7c-1-3p or three putative let-7c-1-3p target mRNAs.

The authors believe that the revised manuscript has improved significantly as a direct result of the review process. Importantly, we are confident that the revised manuscript provides novel insight in several ways, such as revealing that: 1) genetic disruption of the cardiomyocyte circadian clock dysregulates a large number of small RNA species in the heart; and 2) the BMAL1–REV-ERB α/β regulatory network is a mechanism by which the cardiomyocyte circadian clock regulates miRNAs (such as let-7c-1-3p) in the heart.

General Comments for Reviewer 1

“Cardiomyocyte-Specific Disruption of the Circadian BMAL1–REV-ERB α/b Regulatory Network Impacts Distinct miRNA Species in the Murine Heart” by Latimer, Williams, and colleagues works to identify the role that the cardiomyocyte circadian clock (CCC) disruption impacts small RNAs. The group utilizes cardiomyocyte-specific Bmal1 knockout (CBK) mice to identify differentially expressed miRNAs in the CBK mouse. Several of the miRNAs are predicted to be regulated by the REV-ERB α/β . The cardiomyocyte-specific Rev- α/β double knockout mouse (CM-RevDKO) also showed increased expression in several of the miRNAs as the CBK mouse. The authors conclude disruption of the BMAL1-REV-ERV α/β regulatory network can modify miRNAs that influence cell function. This work provides new information beyond the transcriptomic landscape and is expected to significantly impact the field. It opens avenues for future research that facilitate the identification of new mechanisms for circadian regulation of mRNA transcripts, translation, and protein levels.

The authors thank the Reviewer for the complimentary comments.

Specific Comments for Reviewer 1

1) *The authors do a good job listing the limitation in the Discussion section. However, several additional limitations should be included. The study focuses only on male mice, and it remains unclear if this regulatory network exists in female mice.*

We would like to thank the Reviewer for highlighting this limitation. As a direct consequence, the following text has been included in the revised manuscript:

'whether circadian control of miRNA species occurs in either a sex or cell type specific manner remains unknown'

2) Although the authors use cardiomyocyte-specific transgenic models, the tissue analysis contains many different cell types. Depending on the cell type-specific or level of expression for different RNAs, some cardiomyocyte-specific differences in RNA expression might not be resolved. Cell type differences in levels of RNA expression could help explain why only so few small RNA species seemed to be impacted in heart tissue from the transgenic mice. The absolute number of RNA impacted in cardiomyocytes may be much higher.

We would like to thank the Reviewer for highlighting this limitation. As a direct consequence, the following text has been included in the revised manuscript:

'whether circadian control of miRNA species occurs in either a sex or cell type specific manner remains unknown'

3) Although most of the miRNAs explored in the CM-RevDKO mouse do not show a significant time-of-day interaction in the CON or CBK mice, why only test ZT 10 in the CM-RevDKO mice? Especially since there is the time of day that mire-215-5p is likely not expected to be different at this time based on data in Figure 4.

The authors are grateful for the opportunity to clarify why we chose to investigate CM-RevDKO mice at ZT10. We have previously reported that REVERBa and REVERBb protein levels peak in the heart at ZT10. For this reason, we chose to investigate hearts isolated from CM-RevDKO and littermate CON hearts at this time of the day (as we predicted largest differences would be observed at this time of the day).

Accordingly, the following text has been included in the revised manuscript:

'To investigate whether loss of REV-ERB α/β is sufficient to influence expression of the identified putative target miRNA species (independent of BMAL1), let-7c-1-3p, miR-23b-5p, miR-31-5p, miR-139-3p, miR-215-5p, miR-5123, and miR-7068-3p were assessed in cardiomyocyte-specific REV-ERB α/β double knockout (CM-RevDKO) and littermate control hearts collected at ZT10 (a time at which REV-ERB α/β protein levels normally peak in the heart) (39).'

4) Figure 5 Bvi panel has text hidden behind the graph.

Thank you for highlighting this. As a consequence, all figures have been reviewed carefully prior to resubmission.

5) *In the figure legends, “. x-axis shows” Could be “The x-axis...” if one wants to avoid capitalizing the X.” Also, “eight distinct time of the day” could be stated more clearly or make “time” plural.*

Thank you for this suggestion. As a result, all recommended edits have been made to the figure legends.

6) *References are a different font than the main manuscript.*

Thank you for highlighting this discrepancy. The references in the revised manuscript are now in the same font as the main text.

General Comments for Reviewer 2

In this manuscript, Latimer and colleagues aim to assess the extent to which the circadian clock impacts the expression of micro-RNAs in the heart. This study is set in the context that while cardiomyocyte circadian clock (CCC) disruption is known to cause cardiac pathologies, the mechanisms remain poorly understood. The authors postulate that because miRNAs can act as critical modulators of cardiac physiology/pathology they may play a role in the development of CCC disruption disorders. To assess this, the authors performed unbiased small RNA screens in 12-week-old wildtype and cardiomyocyte-specific Bmal1 knockout mice. This revealed a several differentially expressed micro-RNA's, including a number that were under the transcriptional control of REV-ERBa/b. They then used REV-ERBa/b double KO mice to confirm its regulatory control of these small RNA's. Finally, they highlight that one miRNA, let-7c-1-3p, is responsible for the regulation of several genes that are differentially expressed in the CCC disrupted heart. As a result of these observations the authors suggest the exitance of a Bmal1/REV-ERBa/b pathway that regulates miRNA expression prior to the onset of CCC disruption -induced pathology in the heart. Overall, this is a largely descriptive study but it is important as the study is the first of its kind to link circadian disruption to micro-RNA dysregulation in the heart and lays the groundwork for future investigations to help clarify this mechanistically.

The authors thank the Reviewer for the complimentary comments.

Specific Comments for Reviewer 2

1) *As mentioned by the authors, the cardiomyocyte specific Bmal1 KO mice have been previously characterized. It is not clear why in figure 1 they have chosen to add the analysis of additional genes that do not seem to appear anywhere else in the manuscript. In a way, it makes the inclusion of this data unnecessary. Please expand upon why these genes are important or consider removing the data.*

The authors wish to thank the Reviewer for the opportunity to clarify why additional clock components and output genes were assessed. When we previously utilized these samples, we only measured 2 circadian clock components (bmal1, rev-erba) and 1 output gene (dbp) at the mRNA level. Given the importance of convincing the reader that both the clock and clock output are disrupted in these specific samples, we elected

to assess BMAL1 protein levels, as well as 4 clock components (*cry1*, *cry2*, *per1*, *per3*) and 2 clock output genes (*hlf*, *tef*) at the mRNA level. In doing so, we hope that the reader feels confident that the clock and clock output are disrupted in these CBK heart samples (prior to their subsequent use to investigate small RNA species).

Accordingly, the following statements are included in the revised manuscript:

'Previously published studies have confirmed disruption of the circadian clock only in the heart of CBK mice (18, 39). Accordingly, CBK and littermate control (CON) hearts were isolated at 4hr intervals over the 24hr day. Consistent with previously published studies (18), BMAL1 protein levels are decreased by 52.9% at ZT0 in CBK hearts (relative to CON hearts; Figure 1A). We have previously reported attenuated/abolished 24hr rhythms in *bmal1*, *rev-erb α* , and *dbp* mRNA levels in these samples (32); to characterize further the extent of clock disruption in these CBK hearts, additional core clock components (*cry1*, *cry2*, *per1*, *per3*) and clock-controlled genes (*hlf*, *tef*) were investigated at the mRNA level.'

2) *The authors conclude from the data presented in Figure 2 that the main effect on miRNA expression is not time of day dependent, but rather genotype dependent. Importantly though, in both animal models used by the authors they compare KO (cre+/-) to WT (cre-/-) animals. Knowing that Cre-recombinase induces several unrelated effects in the cardiomyocyte, could the authors provide data about Cre-only effects on miRNA expression.*

The Reviewer highlights an important issue. Namely that Cre alone may impact miRNA and/or mRNA levels in the heart. In order to address this issue, new data are included in the revised manuscript, indicating that Cre alone does not affect expression of either let-7c-1-3p, or two putative let-7c-1-3p target mRNAs (*ptpn3*, *usp54*). Although a modest decrease (18.5%) in *dgat2* mRNA levels were observed in MHC α -Cre hearts, the magnitude of this decrease was appreciably less than that observed in CBK (73.7% decrease) and CM-RevDKO (48.1% decrease) hearts at the same/similar time of day. As such, we conclude that Cre alone is not sufficient to account for changes in this important miRNA species.

Accordingly, the following statements are included in the revised manuscript:

'Finally, to investigate the potential off-target contribution of Cre in cardiomyocytes, hearts were collected from MHC α -Cre positive and littermate CON mice at ZT10; RT-PCR analysis revealed no differences in let-7c-1-3p miRNA, *ptpn3* mRNA, and *usp54* mRNA levels between MHC α -Cre and CON mice (Figure 5G). In contrast, a slight (18.5%) decrease in *dgat2* mRNA levels were observed in MHC α -Cre hearts (Figure 5G).'

3) *For Figure 5, including an analysis of the mRNA expression in the CBK mice of the let-7c-1-3p targets would greatly help in demonstrating the importance of this miRNA.*

Thank you for this suggestion. As a direct result, the revised manuscript includes the analysis of 3 putative let-7c-1-3p target mRNAs (*dgat2*, *ptpn3*, *usp54*) in CBK, CM-RevDKO, and littermate CON hearts.

Accordingly, the following statements are included in the revised manuscript:

'RT-PCR and cosinor analysis confirmed that 3 putative let-7c-1-3p targets (*dgat2*, *ptpn3*, and *usp54*) exhibit significant 24hr oscillations in CON at the mRNA level; these oscillations were either abolished (*dgat2*, *ptpn3*) or significantly altered (*usp54*) in CBK hearts (Figure 5E and Supplemental Table 1). Moreover, *dgat2*, *ptpn3*, and *usp54* mRNA levels were decreased in CBK hearts independent of the time-of-day (i.e., genotype main effect; Figure 5E). RT-PCR similarly confirmed decreased expression of *dgat2*, *ptpn3*, and *usp54* mRNA levels in CM-RevDKO hearts at ZT10 (relative to littermate CON hearts; Figure 5F).'

4) *Any mechanistic insights (i.e. mir-mimic's in mice or culture) demonstrating that at least one of the miRNA's like let-7c-1-3p regulate the expression of the particular genes and that this results in some sort of pathology would greatly increase the impact of this work.*

Thank you for this suggestion. Although the authors agree with the Reviewers comments, we believe such assessments are beyond the scope of the current study.

5) *Alternatively, given that the authors chose to study what was happening prior to the onset of any type of CCC disruption-induced pathology, a screen of these miRNA's that demonstrates that they are also enriched in the CBK or REV-ERBa/b DKO diseased hearts would be helpful in demonstrating their importance.*

Thank you for this suggestion. Although the authors agree with the Reviewers comments, we believe such assessments are beyond the scope of the current study.

6) *Can the authors please comment on how the miRNA's in Figure 2 only show a genotype-dependent difference, while their gene targets in Figure 3 show both a time of day and genotype effect?*

Thank you for the opportunity to clarify this. We speculate that clock control of miRNA species is important for setting daily average levels of mRNAs (and potentially protein levels as well), whereas other clock-controlled mechanisms are involved in establishing 24hr oscillations. Consider *dgat2*, for example. We have previously reported that *dgat2* is a direct target of both the BMAL1/CLOCK heterodimer, as well as the clock-controlled gene E4BP4 (references 18 and 40, respectively); changes in the activity of these transcription factors likely drives 24hr oscillations in *dgat2* mRNA levels in the heart. In the current study, our observations are consistent with an important role of let-7c-1-3p for setting the daily average level of *dgat2* mRNA.

7) *Blots provided in supplemental data appear to have a lot of non-specific binding for*

Bmal1, with the lowest intensity band being indicated as *Bmal1*. Please provide a positive control.

Thank you for this suggestion. We have previously reported that in cardiomyocytes isolated from CBK hearts, the band at the correct molecular weight for BMAL1 decreases by >90% (reference 18). We consider this as a positive control.

8) *The representative blots shown in Figure 1Ai and 3 Ci should be improved.*

Thank you for the opportunity to clarify. After semi-quantification of the western blots through densitometry, we calculated group average values (for CBK and CON). We next reviewed individual sample densitometry values, to identify which lanes/samples were truly representative of the group average (i.e., closest to the group average value). Based on this methodology, we chose the lanes shown in the representative blots.